# Preparation and Application of Double Network Interpenetrating Colon Targeting Hydrogel Based on Konjac Glucomannan and N-Isopropylacrylamide

**DOI:** 10.3390/gels9030221

**Published:** 2023-03-14

**Authors:** Renhua Yao, Xiaoqin Yu, Rui Deng, Huarong Zou, Qingwen He, Wenfeng Huang, Chunxiao Li, Kun Zou

**Affiliations:** 1Hubei Key Laboratory of Natural Products Research and Development, College of Biological and Pharmaceutical Sciences, China Three Gorges University, Yichang 443002, China; 2Hubei Hongyu New Packing Material Co., Ltd., 1 Juxiang Avenue, Jiaqueling Town, Yiling District, Yichang 443000, China; 3School of Health Care and Nursing, Hubei Three Gorges Polytechnic, Yichang 443000, China

**Keywords:** konjac glucomannan, N-isopropylacrylamide, double network interpenetrating hydrogel, colon targeting

## Abstract

Konjac glucomannan (KGM) can be degraded by colon-specific enzymes in the colonic environment, making it one of the materials for treating colonic diseases, which has attracted more and more attention. However, during drug administration, especially in the gastric environment and due to its easy swelling, the structure of KGM is usually destroyed and the drug is released, thereby reducing the bioavailability of the drug. To solve this problem, the easy swelling and drug release properties of KGM hydrogels are avoided by creating interpenetrating polymer network hydrogels. In this study, N-isopropylacrylamide (NIPAM) is first formed into a hydrogel framework under the action of a cross-linking agent to stabilize the gel shape before the gel is heated under alkaline conditions to make KGM molecules wrap around the NIPAM framework. The structure of the IPN(KGM/NIPAM) gel was confirmed using Fourier transform infrared spectroscopy (FT-IR) and x-ray diffractometer (XRD). In the stomach and small intestine, it was found that the release rate and swelling rate of the gel were 30% and 100%, which were lower than 60% and 180% of KGM gel. The experimental results showed that this double network hydrogel has a good colon-directed release profile and fine drug carrier ability. This provides a new idea for the development of konjac glucomannan colon-targeting hydrogel.

## 1. Introduction

Konjac glucomannan (KGM) is a non-ionic water-soluble polysaccharide composed of D-glucose and D-mannose through β-1, 4, and 4 pyranoside bonds. It is mainly derived from konjac tubers and is a kind of abundant natural renewable resource [1,2,3,4]. KGM cannot be degraded in the upper gastrointestinal tract but can only be degraded by β-mannanase or β-glucosidase in the colon. Based on this characteristic, KGM is an ideal material for preparing colon-targeted preparations [5,6,7,8]. However, konjac glucomannan is easy to swell, causing its gel structure to be easily destroyed in the upper gastrointestinal tract, especially in the stomach environment, resulting in drug release. The effective drug concentration cannot be reached at the lesion site, reducing bioavailability [9,10,11].

Many scholars have proposed different solutions to this problem [12,13,14]. At present, the commonly used solution is to covalently modify KGM to reduce its swelling property and make the hydrogel more stable in the upper digestive tract environment, see Liu et al. [15] Sodium phosphate chemically modified KGM shielded the electronic strength of negative charges by increasing the ionic strength to reduce the swelling ratio. Liu et al. [16] copolymerized KGM with acrylic acid to adjust the swelling degree and pH sensitivity of the gel by changing the cross-link density of the polymer or cross-linking KGM with the help of cross-linking agents or cations (such as Ca^2+^) in a physical blending manner. Zou [17] and colleagues used materials such as KGM and N-isopropylacrylamide, graphene oxide (SA), physical blending, and physical cross-linking with the help of Ca^2+^, prepared a compound with both temperature sensitivity and good mechanical properties. Ding et al. [18] modified konjac glucomannan (KGM) with phenylalanine (Phe), and hydrogels were formed and mediated using the cross-linker cucurbit uril (CB [8]. Although these methods have achieved good results, there are some limitations. First, KGM hydrogels prepared via irreversible covalent cross-linking usually maintain the properties of the gel through covalent bonds. The preparation process and the introduction of a large amount of impurities, secondly, lengthy and time-consuming optimization of conjugation conditions is required when adjusting the degree of cross-linking, and thirdly, this method usually has a slow release rate, which is not conducive to drug release.

Interpenetrating Polymer Network (IPN) hydrogel refers to a hydrogel network interspersed in a hydrogel system formed by another or several networks in the form of noncovalent cross-linking, which not only has the properties shared by the two networks but also is easy to synthesize [19,20]. Yao [21] prepared double-network interpenetrating hydrogels by drying konjac glucomannan (KGM) hydrogels and then polymerizing them with acrylic acid. However, this method will form large pores between acrylic acid and konjac glucomannan during polymerization, which cannot wrap the drug well, the drug loading is low, and the drug release is slow.

Poly(N-isopropylacrylamide) (PNIPAM) is a temperature-sensitive material [22]. It is synthesized through the free radical polymerization of monomer isopropylacrylamide, and its polymerized PNIPAM molecular side chain has both a hydrophilic amide group (–CONH–) and a hydrophobic isopropyl group [–CH(CH_2_)_3_] [23,24]. When the temperature rises above the critical temperature (LCST) (about 31 °C) [25], the hydrogen bond between the amide group and the water molecule is destroyed, and the hydrophobicity of the isopropyl group in the structure plays a dominant role, making the material hydrophobic [26,27,28].

Fluorouracil (5-FU) has always been the first-line drug for the clinical treatment of colon cancer. It inhibits DNA synthesis in cancer cells by inhibiting thionine synthase in cancer cells. At the same time, 5-FU can be combined into the RNA of cancer cells, eventually leading to the apoptosis of cancer cells [29,30]. Fluorouracil is usually given intravenously, which is often accompanied by inflammation of the vein. The main disadvantage of oral administration is gastrointestinal reactions and low bioavailability, so colon-targeted preparations are a good solution. This is consistent with the intention of this study to prepare a colon-targeted release drug gel formulation, so this study uses 5-FU as a model drug.

In view of the above facts, we developed the KGM-NIPAM double interpenetrating network hydrogel here. Through the deacetylation of konjac glucomannan with sodium carbonate, the KGM molecular chains were interwoven to form a network and entangled in the poly(N-isopropylacrylamide) as the backbone of the gel system, together constituting an interpenetrating double network hydrogel. The hydrogel is noncovalently cross-linked with KGM and NIPAM to form a double network interpenetrating structure, and it is applied to the colon-localized release of 5-FU. In the face of current polysaccharide-based hydrogels, they are unsuitable for most preclinical applications due to their limited mechanical properties and poor tunability [31,32]. The IPN gel structure proposed in this topic can be prepared by changing the skeleton material of the IPN gel network to meet the different requirements of polysaccharide-based gels. This can provide new research ideas for the preparation and optimization of polysaccharide-based hydrogels for clinical applications.

## 2. Results and Discussion

### 2.1. Synthesis of IPN (KGM/NIPAM) Hydrogel

IPN (KGM/NIPAM) hydrogel was prepared using IPN technology. Since the gel was mainly composed of KGM and NIPAM, to explore the effect of the ratio of KGM and NIPAM on the formation of the gel, this paper studied 3 different KGM and NIPAM hydrogels. The ratios are listed in Table 1 below. As shown in Figure 1, in the inverted bottle experiment, within the same gelation time, the surface of gel A was flat, the surface of gel B was slightly exuded, and the surface of gel C was uneven. It shows that the higher the NIPAM content, the better the gel morphology. This was because the network formed by the NIPAM gel acts as a skeleton in the IPN hydrogel. When the NIPAM ratio has a higher value and the framework is denser, the shape of the gel supported by the framework is better.

**Table 1 gels-09-00221-t001:** Feed ratio scheme for formulating IPN (KGM/NIPAM) hydrogel.

Sample	m(KGM):m(NIPAM)	m(MBA):m(NIPAM)	m(APS):m(NIPAM)	v(TEMED):m(NIPAM)
A	0.5:1	3:100	1:100	1:1000
B	1:1	3:100	1:100	1:1000
C	2:1	3:100	1:100	1:1000

### 2.2. Scanning Electron Microscope (SEM)

The surface morphology of KGM, NIPAM, and IPN hydrogels frozen using a freeze-dryer was characterized by SEM, as shown in Figure 2a–c. It can be observed from image A that the inside of KGM hydrogel is arranged in sheets, and the pieces are intertwined. There are many large pores that do not seem to be tight, which will lead to easy drug release, which is not conducive to drug protection. It was observed from image b that NIPAM hydrogel was closely bonded inside, like a whole, with small pores and a small amount, which would help protect the drug from being destroyed by the external environment but would make it difficult to load the drug. It can be observed from image c that IPN hydrogel presents a porous network structure with small pores and a large number of pores, which is conducive to drug loading. Meanwhile, the entire gel looks interwoven like a whole, which is conducive to the stability of gel morphology that protects drugs.

### 2.3. Structure Characterization

The Fourier-transform infrared spectroscopy (FT-IR) of KGM hydrogel, NIPAM hydrogel, and IPN (KGM-NIPAM) are shown in Figure 3a. In the figure, the wave number 1636 cm^−1^ represents the associated amide bond, and the N-H bending vibration and C–N stretching vibration are represented by 1543 cm^−1^, characteristic peaks of PNIPAM; 805 cm^−1^ represents the pyran ring and β-glucoside bond absorption peak. It can be seen from the figure that the characteristic peaks of PNIPAM and KGM gels are all reflected in the IPN hydrogels. At the same time, due to the interaction of different groups, the infrared absorption peaks of IPN hydrogels overlap, resulting in some strong and some weak peaks, proving the successful synthesis of IPN hydrogels [17].

Figure 3b shows the XRD patterns of NIPAM hydrogel, KGM hydrogel, and IPN hydrogel. In the XRD pattern of NIPAM gel, it is observed that there is a fundamental peak at 2θ = 21.7° and a secondary peak at 2θ = 8.4°, which is the semi-crystalline structure characteristic of P(NIPAM) copolymer. This is comparable to the XRD patterns of NIPAM gels obtained by other workers [33]. A broad peak at 2θ = 21.3° can be observed from KGM gel, indicating that KGM is an amorphous material, which was consistent with the XRD pattern of KGM gel obtained by other workers [14]. In the XRD pattern of IPN gel, it can be seen that the gel has a secondary peak at 2θ = 8.2° and a broad peak near 2θ = 21°, which indicates that there are both KGM gel components and the composition of NIPAM gel. Moreover, it was found that IPN gel had some sharp peaks at 2θ = 31.9° and 2θ = 45.7°, indicating that NIPAM crystals were produced [34]. This indicated that KGM interacted with NIPAM in the preparation of IPN hydrogel.

### 2.4. Thermosensitivity and Uptake of IPN Hydrogen

As shown in Figure 4a, in IPN gel, with the change of time and pH value, the gel will uptake rapidly in simulated gastric fluid (0~2 h, pH1) and simulated intestinal fluid (3~5 h, pH 6.8). When we simulated a colonic environment (6~12 h, pH 7.4), the gel uptake rate slowed. The gel uptake rate and simulated intestinal fluid environment did not change much, and it was found that the higher the proportion of NIPAM in all gels, the slower the uptake rate of the gel. This is due to the hydrophobic effect of its polymer chains, which limits the penetration of external water molecules into the gel, and the skeleton formed by NIPAM at body temperature limits the uptake of KGM inside the gel. As time increases, water molecules enter, and KGM gradually uptake, constantly resisting the bondage of the NIPAM skeleton, thereby delaying the uptake process. When the proportion of NIPAM is higher, the density of hydrophobic groups is also greater, so the uptake rate is smaller. Figure 4b shows the change in the uptake rate of the same gel in an aqueous solution of pH 7 at different temperatures. It can be seen from the figure that at room temperature (25 °C), the uptake degree of the gel is significantly higher than that at higher temperature (37 °C). This is mainly due to the critical temperature of NIPAM being around 31 °C. When the temperature is lower than the critical temperature, the gel molecules have good hydrophilicity and a high uptake rate; when the temperature is higher than the critical temperature, the gel molecules are highly hydrophobic, and the uptake rate is low.

### 2.5. Drug Loading of IPN Hydrogel

As shown in Figure 5, different proportions of IPN hydrogels have different drug loadings. When KGM: NIPAM is 1:1, the maximum drug loading of the gel is 35%, and when KGM: NIPAM is 1:1, the lower drug loading rate is about 10%.

### 2.6. In Vitro Release of IPN Hydrogel

In Figure 6, we simulate the release process of the gel in the human digestive tract (stomach, small intestine, colon) and detect the drug release process using an ultraviolet spectrophotometer. From the figure, we can see that the pure KGM hydrogel shows obvious release in the stomach and small intestine; at 6 h in vivo, about 90% of the drug is released and remains unchanged. While the IPN hydrogel release rate in the stomach environment is less than 30%, and in the subsequent colon environment, with the participation of enzymes, the release rate is significantly improved. This is because the skeleton formed by NIPAM inhibits the swelling tendency of KGM in the gel, so that the gel’s smaller pores are not conducive to drug release. At the same time, different proportions of KGM have different release rates over 6–12 h. Due to the presence of β-glucosidase in the environment of the colon, under the action of β-glucosidase, the glycosidic bond in KGM is broken by the enzyme, and the KGM chain breaks, leading to the expansion of gel pores and promoting the release of 5-FU.

### 2.7. In Vivo Administration of IPN Gel

As shown in Figure 7a, we found that direct administration of an 5-FU aqueous solution to the stomach could cause the plasma of mice to increase rapidly within 2 h and reach a peak, then decrease rapidly, and no drug could be detected about 12 h later. But taking a potion-loaded gel orally slows this process down significantly, peaking around 10 h, and blood levels of the drug didn't wane quickly but persisted for some time. As shown in Figure 7b, in the contents of the small intestine, the 5-FU aqueous solution group reached maximum concentration about 2 h later and then gradually decreased, while the gel group was always detected and maintained at a certain concentration. As can be seen from Figure 7c, in the colon contents, the 5-FU aqueous solution group was unable to detect drugs after 8 h. In comparison, the drug-loading gel group showed obvious concentration changes around 12 h, indicating that IPN hydrogel did have the effect of colon-targeted drug release. In the experiment, it was found that due to the individual differences and the disadvantages of the oral administration method of mice, the blood drug concentration of the tested mice at the same time point was significantly different.

## 3. Conclusions

In this study, we explored the feasibility of konjac glucomannan as a colon-targeted drug delivery system. The IPN (KGM-PNIPAM) hydrogel has strong hydrophilicity and a high swelling rate under a low-temperature environment, which benefits the drug loading in the gel. In the human environment (37 °C), the isopropyl group in the PNIPAM structure plays a hydrophobic role. The NIPAM as the backbone limits the swelling of KGM, thereby protecting the drug from being released in the gastrointestinal environment. However, when it comes to the colonic environment, under the action of glycosidase, the KGM network is destroyed by the enzyme, making the gel porous, which is conducive to the release of the drug. In the in vitro and in vivo release experiments of the hydrogel loaded with 5-FU, it can be seen that the drug content changes with time, and the effect of delaying the release of the drug is obvious. Therefore, this method can provide a new idea for realizing the research based on KGM as a colon-targeting agent.

## 4. Materials and Methods

### 4.1. Materials

Konjac glucomannan (KGM, Xuanqinrun Trading Co., Ltd., >95%, Yichang, China), N-isopropylacrylamide (NIPAM), N, N′-Methylenebisacrylamide (MBA), Ammonium persulphate, Mannanase (CAS: 37288-54-3), N, N, N′, N′-Tetramethylethylenediamine (TEMED) and other reagents were of analytical grade, purchased from Shanghai Macklin Biochemical Technology Co., Ltd., Shanghai, China and all experimental water was purified.

### 4.2. Preparation of KGM, NIPAM and IPN (KGM/NIPAM) Hydrogel

KGM Hydrogel: The purified KGM powder was put into a container and dissolved by stirring in distilled water in a certain proportion. Add an appropriate amount of sodium carbonate solution to the KGM aqueous solution and then heating it in a 65 °C water bath for 1 h. Rinse with water 3 times (1 h each time), dry and weigh ( Figure 8a).

NIPAM Hydrogel: Put NIPAM powder into a container and dissolve it in distilled water in a certain proportion. The crosslinking agent N-isopropylacrylamide and ammonium persulfate were added to the mixture. Stir the mixture under nitrogen for 10 minutes, then add the TEMED with a syringe. The prepared NIPAM polymer hydrogel was rinsed with water for 3 times (1 h each time), dried and weighed ( Figure 8b).

IPN (KGM/NIPAM) Hydrogel: The purified KGM powder and NIPAM powder were put into a container and dissolved in distilled water in a certain proportion. The cross-linking agent N-isopropylacrylamide and ammonium persulfate were added to the mixture. The mixture was stirred under nitrogen for 10 min, then TEMED was added with a syringe. The prepared NIPAM copolymer hydrogel was soaked in sodium carbonate solution for 10 min, then heated in a 65 °C water bath for 1 h, washed with water 3 times (1 h each time), dried, and weighed (Figure 8c).

### 4.3. Preparation of IPN (KGM/NIPAM) Drug-Loaded Hydrogel

A certain amount of fluorouracil was weighed and dissolved in distilled water. The dried IPN (KGM/NIPAM) hydrogel was completely immersed in it, swelled at room temperature for 4 h, and washed with distilled water several times to remove the drug on the surface of the gel. The prepared gel was refrigerated for later use.

### 4.4. Structure Characterization

The internal morphology of the gel was measured using a scanning electron microscope (SEM, Japan Electronics, Tokyo, Japan, JSM-7500F). First, gold was spray-coated onto the surface of the lyophilized hydrogel, and then SEM images were obtained using electron beam scanning at an accelerating voltage of 15 KV.

KGM hydrogels, PNIPAM hydrogels, and IPN double network hydrogels were detected using Fourier transform infrared spectroscopy (FTIR, USA PE, Waltham, Mass, USA, Fourier NIR). The FTIR spectra of the samples were acquired through scanning in the range of 4000–600 cm^−1^. All three gels remain dry during the measurement.

X-ray diffraction detection (XRD, Nisei, Kobe, Japan, Ultima IV). First, KGM hydrogel, NIPAM hydrogel, and IPN hydrogel were dried at 45 °C. Then XRD tests were conducted in the range of 10°~80° with copper as the target material and an X-ray diffraction power of 3 kw.

### 4.5. Thermosensitivity and Uptake of IPN Hydrogel

A certain amount of dry IPN hydrogel (M_0_) was accurately weighed, and the dry hydrogel was put into 37 °C strong acid solution (0.1 mol/L hydrochloric acid) for 2 h, weak acid solution (pH 6.8) for 4 h and weak basic solution (pH 7.4) for 6 h. To simulate the uptake of IPN hydrogel in the human digestive tract (stomach, small intestine, colon) environment. Take out IPN hydrogel at regular intervals, blot the excess water on the surface with filter paper, weigh it to get (M_S_), and calculate its uptake rate according to the following formula:[Uptake ratio] = (M_S_ − M_0_)/M_0_(1)

### 4.6. Study on Temperature Sensitivity of IPN Hydrogel

Take a certain mass of dry hydrogel (Md), put it in 300 mL PBS aqueous solution, observe the mass (Mw) of IPN hydrogel at different time intervals and under different temperature conditions, and calculate the rate of change according to the following formula:[Uptake ratio] = (Mw − Md)/Md(2)

### 4.7. Drug Loading

Precisely weigh a certain amount of dry IPN (KGM/NIPAM) hydrogel (M_S_), completely immerse in a certain concentration of 5-FU aqueous solution, uptake at room temperature for 4 h, rinse with distilled water several times to remove the drug on the surface of the gel, freeze-dry, weigh amount (Mt), and use the following formula to calculate the drug loading rate:[Drug loading rate] = (Mt − M_S_)/M_S_(3)

### 4.8. Study of Drug Release from IPN Hydrogel In Vitro

A certain mass of drug-loaded IPN hydrogel was placed in a dialysis bag filled with 5 ml of PBS. The dialysis bag was placed in 300 mL of an acidic solution (pH 1.2) for 2 h to simulate the gastric environment and a neutral solution (pH 6.8) for 4 h to simulate the intestinal environment. A weak alkaline solution (pH 7.4) and a certain amount of mannanase were added to the dialysis bag for 6 h to simulate the drug release of IPN hydrogel in the human colon environment condition. The solution is withdrawn at predetermined intervals and replaced with a fresh solution. Fluorouracil concentrations were calculated using ultraviolet-visible spectrophotometry (UV-Vis, Shanghai Yuan Analysis Instrument Co., LTD., Shanghai, China, UV-5900).

### 4.9. In Vivo Administration of IPN Gel

The experimental mice (male mice, Kunming species, 20–24 g, provided by the Animal House of China Three Gorges University, Yichang, China) were randomly divided into 2 groups, with 12 mice in each group. They fasted for 12 h and drank water freely before taking medicine.

The control group was given a 5-fu aqueous solution (dose 10 mg/kg) through intragastric administration. The experimental group was given IPN-loaded potion gel (the overall dose was 10 mg/kg) via gavage. After intragastric administration, mouse plasma, small intestine contents, and colon contents were collected at 1, 2, 4, 6, 8, 10, 12, 16, 20, and 24 h, respectively.

The plasma stood for 2 h, then was centrifuged, the upper serum was taken, the upper serum was added with ethyl acetate and shaken for 30 min, the upper solvent was taken, the upper solvent was dried, and the mobile phase was added. The content of 5-fu was measured using HPLC (HPLC, Shimazu Production House, Kyoto, Japan, LC-2030C 3D). The contents of the small intestine and colon were ground, shaken for 30 minutes after adding ethyl acetate, and the upper solvent was taken. After drying, the mobile phase was added, and the content of 5-fu was measured using HPLC [35].

### 4.10. Statistical Analysis

Statistical evaluations of data were implemented 3 times. Analysis of variance (ANOVA) was used to identify the statistical significance between different groups. All results were expressed as mean ± standard deviation (SD). In all studies, *p* < 0.05 was considered to have statistical significance.

## Figures and Tables

**Figure 1 gels-09-00221-f001:**
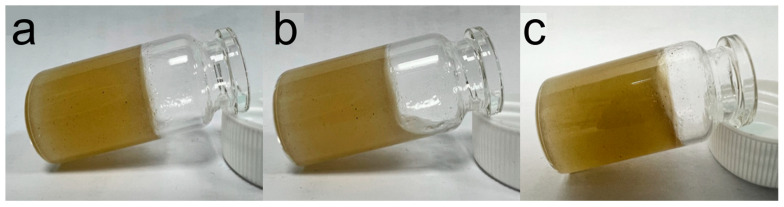
(**a**–**c**) three different ratio gel inversion experiments.

**Figure 2 gels-09-00221-f002:**
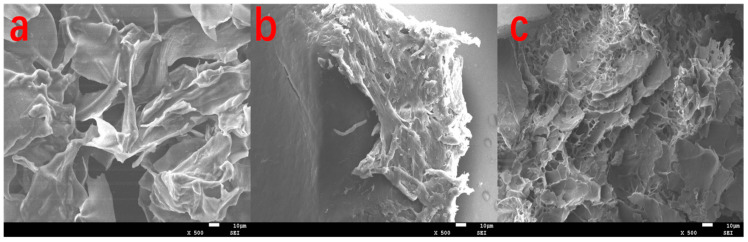
SEM images from KGM hydrogel (**a**), NIPAM Hydrogel (**b**), and IPN hydrogel (**c**).

**Figure 3 gels-09-00221-f003:**
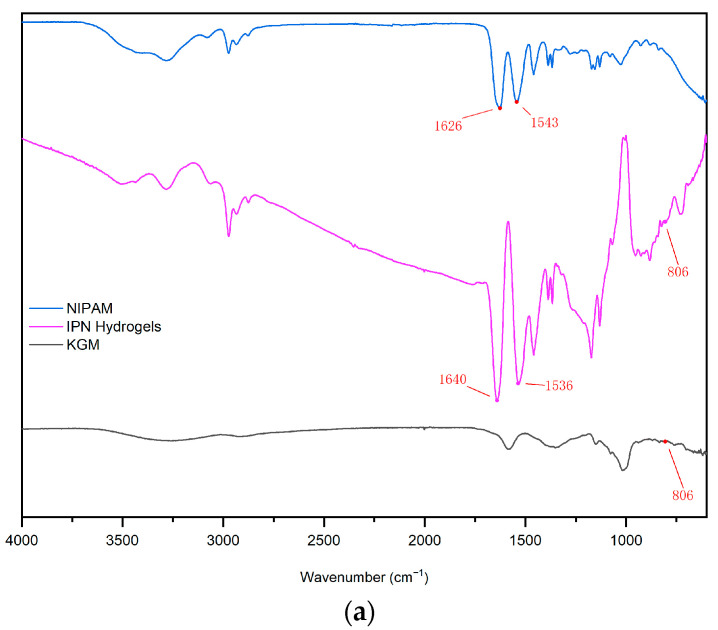
(**a**) The FT-IR spectra of KGM gel, NIPAM gel and IPN gel; (**b**) The XRD spectra of KGM gel, NIPAM gel and IPN gel.

**Figure 4 gels-09-00221-f004:**
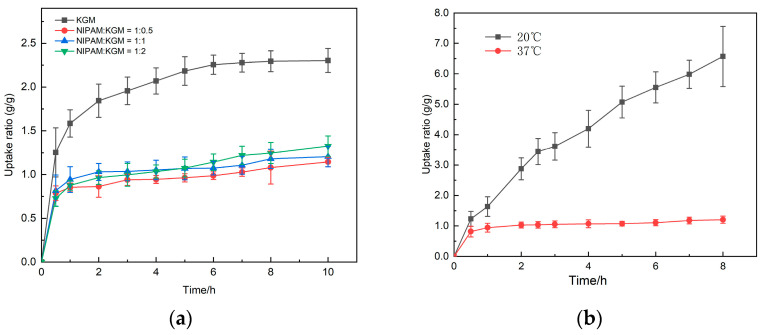
(**a**) The picture shows the swelling of hydrogels with different ratios under the temperature and pH conditions that simulate the environment in the human body (37 °C, 0–2 h, 0.1 mol/L hydrochloric acid solution; 3–6 h, pH6.8; 7–10 h, pH7.4); (**b**) The picture shows the swelling of IPN hydrogel (KGM: NIPAM = 1:1) at different temperatures.

**Figure 5 gels-09-00221-f005:**
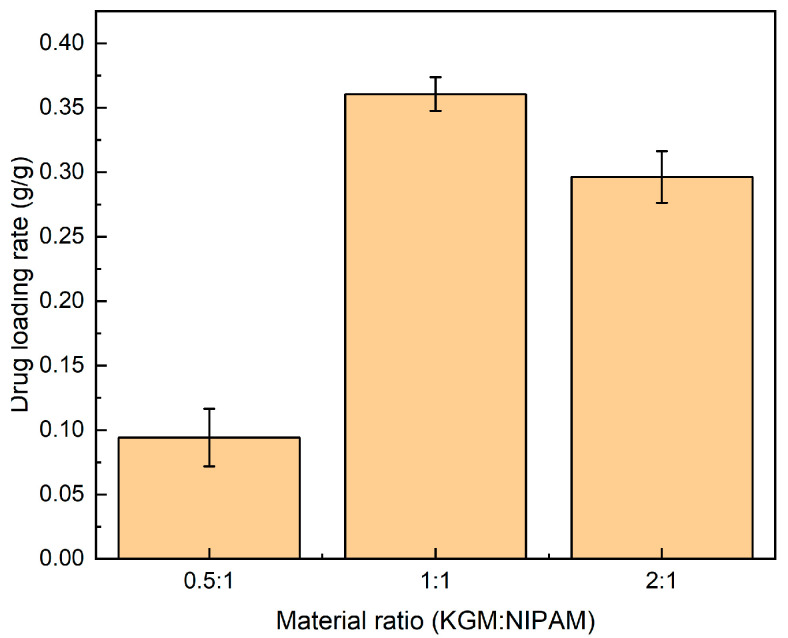
This picture shows the drug loading of hydrogels with different ratios under the same conditions.

**Figure 6 gels-09-00221-f006:**
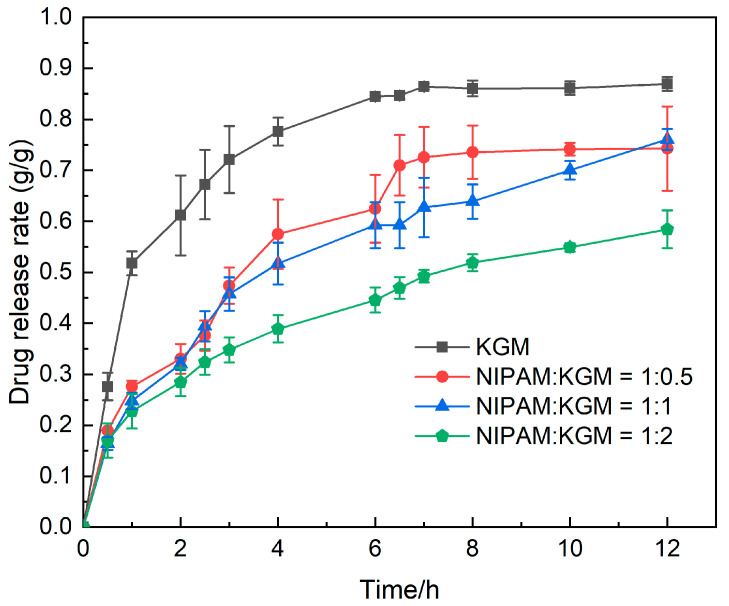
This picture shows the drug(5-FU) release of hydrogels with different ratios in vitro under simulated human internal environment conditions (37 °C, 0–2 h, 0.1 mol/L hydrochloric acid solution; 3–6 h, pH6.8; 7–12 h, pH7.4, Mannanase).

**Figure 7 gels-09-00221-f007:**
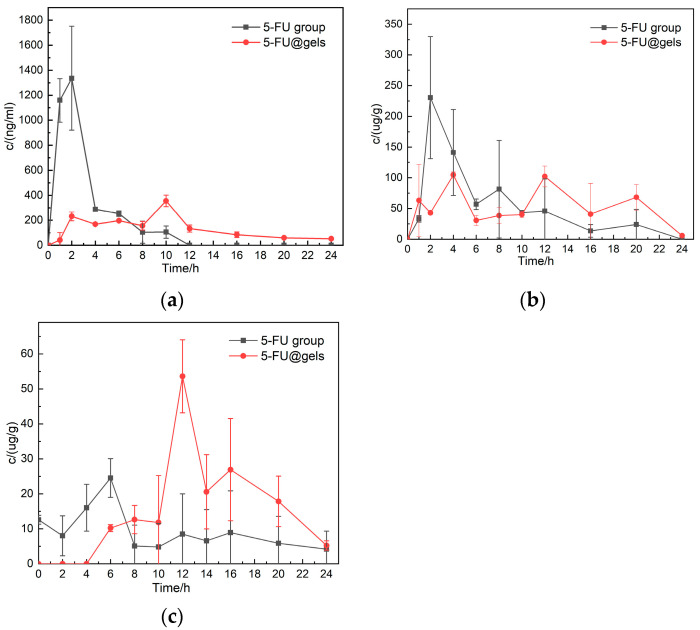
In vivo changes of 5-FU released by oral intragastric administration of 5-FU and 5-FU@gels ((**a**); Serum, unit: ng/mL. (**b**); Contents of small intestine, unit: ug/g. (**c**); Colon contents, unit: ug/g).

**Figure 8 gels-09-00221-f008:**
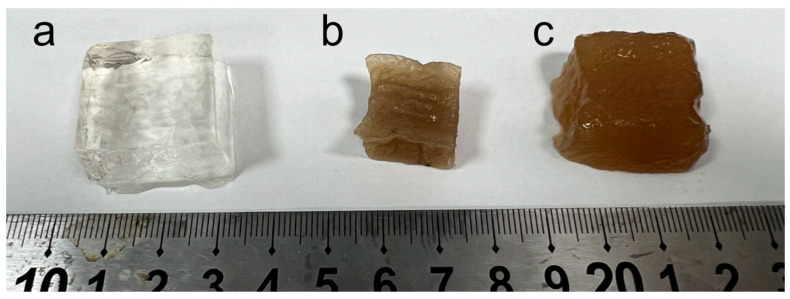
(**a**) shows NIPAM hydrogel; (**b**) shows IPN (KGM/NIPAM) hydrogel; (**c**) shows KGM hydrogel.

## Data Availability

Not applicable.

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
