# Peer review of "Preparation and Application of Double Network Interpenetrating Colon Targeting Hydrogel Based on Konjac Glucomannan and N-Isopropylacrylamide"

_gels, 2023, doi:10.3390/gels9030221_

Round 1
Reviewer 1 Report
The paper reports on hybrid network made of natural and synthetic polymers for colon-targeted drug release. The paper could be of interest, but it requires major revisions. The Abstract should be improved by adding the quantitative results found by the authors. Please, avoid generalities and claims that are not directly proved by results. All the acronyms used in the manuscript must be explained before reported in the text. In the paper IPN is used and never clarified. The authors claim about FTIR characterization which could prove the formation of the composite hydrogel. I don't think that FTIR is the best technique to prove the formation of polymer network. The spectra reported in Figure 3 (that must have the Y-axis labeled with the quantity reported together with the unit of measure) only show the presence of same chemical groups in the mixture of the two polymer precursors but it does not prove how the mixture is cross-linked. I suggest the authors to further investigate the nature of the resulting hydrogel by other techniques such as XRD, TGA, DSC and find features that could be ascribed to the formation a new polymers network. Please, check the Y-axes of Figure 5 and 6 and add the unit of measure. In paragraph 2.5, reading the text it seems that authors report in vivo experiments, whereas they declared that they only performed in vitro ones. Please, check this point. Why did the authors used 5-FU? Which is its action? Is a hydrophobic or a hydrophilic drug? Please, better explain this point.
Author Response
Dear Reviewer,
Thank you for your suggestions on our manuscript. Those comments are all valuable and very helpful for revising and improving our paper, as well as the important guiding significance to our researches. We have studied comments carefully and have made correction which we hope meet with approval. Revised portion are marked in blue in the paper. The main corrections in the paper and the responds to the reviewer’s comments are as flowing:
Point 1: “The Abstract should be improved by adding the quantitative results found by the authors. Please, avoid generalities and claims that are not directly proved by results. All the acronyms used in the manuscript must be explained before reported in the text.”
Replying: Yes, thanks for your kind suggestion. I have added the following sentences to the text:
In the environment of stomach and small intestine, it was found that the release rate and swelling rate of the gel were 30% and 100% which were lower than 60% and 180% of KGM gel.
Interpenetrating Polymer Network (IPN) hydrogel refers to a hydrogel network interspersed in a hydrogel system formed by another or several networks in the form of non-covalent crosslinking which not only has the properties shared by the two net-works, but also is easy to synthesize. Yao prepared double-network inter-penetrating hydrogels by drying konjac glucomannan (KGM) hydrogels and then polymerizing them with acrylic acid. However, this method will form large pores between acrylic acid and konjac glucomannan during polymerization, which cannot wrap the drug well, and the drug loading is low and the drug release is slow.
Point 2: “I don't think that FTIR is the best technique to prove the formation of polymer network. I suggest the authors to further investigate the nature of the resulting hydrogel by other techniques such as XRD, TGA, DSC and find features that could be ascribed to the formation a new polymers network.”
Response 2: To solve this problem, We added SEM and XRD experiments in the part of gel characterization. KGM gel has porous structure but loose structure, NIPAM gel has dense structure but few pores, IPN gel is dense and porous, which has the advantages of KGM gel and NIPAM gel. In XRD, the IPN gel pattern was similar to KGM gel and NIPAM gel in peak shape, and NIPAM crystal was formed, indicating that NIPAM and KGM did interact. Combined with SEM, FT-IR and XRD patterns, IPN gel was composed of KGM gel and NIPAM gel, and the two components interacted with each other. The product had the advantages of the two gels and was in line with the characteristics of IPN gel.
Point 3:“ In paragraph 2.5, reading the text it seems that authors report in vivo experiments, whereas they declared that they only performed in vitro ones. Please, check this point. Why did the authors used 5-FU? Which is its action? Is a hydrophobic or a hydrophilic drug?”
Response 3: To solve this problem, We added in vivo drug release experiment, and tested the content of 5-FU in the plasma, small intestine contents and colon contents of mice after oral intragastric administration. The function of 5-FU is explained in the introduction:
Fluorouracil(5-FU) has always been the first-line drug for the clinical treatment of colon cancer. It inhibits the synthesis of DNA in cancer cells by inhibiting thionine synthase in cancer cells. At the same time, 5-FU can be combined into RNA of cancer cells, eventually leading to apoptosis of cancer cells. Fluorouracil is usually given intravenously, which is often accompanied by inflammation of the vein. The main disadvantage of oral administration is gastrointestinal reactions and low bioavailabil-ity, so colon-targeted preparations are a good solution. This is consistent with the in-tention of this study to prepare a colon-targeted release drug gel formulation, so this study uses 5-FU as a model drug.

Reviewer 2 Report
1. These elements (KGM, and NIPAM) added by the authors in the material design process and their respective advantages should be explained. In this way, the reader will have a clearer idea of the features of this article.
2. The manuscript lacks any information on experimental replication. This is particularly worrisome. Please revise the manuscript detailing your experimental and technical replications (refering 10.1016/j.cej.2023.141852)
3. The "swelling" refers to the change in volume and is the ratio of the new volume to the old volume (in mL/mL). The increase in mass divided by the original mass is the "uptake" (in g/g). Unfortunately, the misuse of this terminology is very common.
4. Where are these hydrogels going to be used in real life? Advantages of the designed IPNs can be improved by comparing and citing 10.1016/j.carbpol.2020.117213; 10.1021/acsami.9b20794. The novelty of this work can be described at the end of the Introduction. It could be better if a brief comment (challenges and future prospects) is added at the manuscript.
5. There are some formatting errors in the article. For example, spelling of references must be checked to meet the journal style (such as Reference 13). Kindly check for correctness.
Author Response
Dear Reviewer,
Thank you for your suggestions on our manuscript. Those comments are all valuable and very helpful for revising and improving our paper, as well as the important guiding significance to our researches. We have studied comments carefully and have made correction which we hope meet with approval. Revised portion are marked in blue in the paper. The main corrections in the paper and the responds to the reviewer’s comments are as flowing:
Point 1: These elements (KGM, and NIPAM) added by the authors in the material design process and their respective advantages should be explained. In this way, the reader will have a clearer idea of the features of this article.
Replying: I explained in the preface that KGM can be degraded by mannanase, which only exists in the colon, indicating that KGM is a good colon-targeted drug delivery material. When NIPAM was above the critical temperature (31℃), the hydrophobicity was enhanced, and when NIPAM was below the critical temperature, the hydrophilicity was enhanced, which was conducive to the drug downloading at room temperature (5-FU dissolved in water), and inhibited the release of the drug in human environment.
Point 2: The manuscript lacks any information on experimental replication. This is particularly worrisome. Please revise the manuscript detailing your experimental and technical replications
Response 2: All of my experimental data are reproducible, which I have explained in the article.
Statistical evaluations of data were implemented 3 times. Analysis of variance (ANOVA) was used to identify the statistical significance between different groups. All results were expressed as mean ± standard deviation (SD). In all studies, p < 0.05 considered to be statistical significance.
Point 3: The "swelling" refers to the change in volume and is the ratio of the new volume to the old volume (in mL/mL). The increase in mass divided by the original mass is the "uptake" (in g/g). Unfortunately, the misuse of this terminology is very common.
Response 3: This issue has been changed in the article.
Point 4: Where are these hydrogels going to be used in real life? Advantages of the designed IPNs can be improved by comparing and citing 10.1016/j.carbpol.2020.117213; 10.1021/acsami.9b20794. The novelty of this work can be described at the end of the Introduction. It could be better if a brief comment (challenges and future prospects) is added at the manuscript.
Response 4: This issue has been changed in the article.
In the face of current polysaccharide-based hydrogels, they are not suitable for most preclinical applications due to their limited mechanical properties and poor tunability. The IPN gel structure proposed in this topic can be prepared by changing the skeleton material of the IPN gel network to meet different requirements of polysaccharide-based gels. This can provide new research ideas for the preparation and optimization of polysaccharide-based hydrogels for clinical applications.
Point 5: There are some formatting errors in the article. For example, spelling of references must be checked to meet the journal style (such as Reference 13). Kindly check for correctness.
Response 5: This issue has been changed in the article.

Round 2
Reviewer 1 Report
The revised paper can be published.